# Moisture Content and Mechanical Properties of Bio-Waste Pellets for Fuel and/or Water Remediation Applications

Yuriy A. Anisimov [1], Bernd G. K. Steiger [1], Duncan E. Cree [2,*] and Lee D. Wilson [1,*]

1 Department of Chemistry, University of Saskatchewan, Saskatoon, SK S7N 5C9, Canada
2 Department of Mechanical Engineering, University of Saskatchewan, Saskatoon, SK S7N 5A9, Canada
* Correspondence: duncan.cree@usask.ca (D.E.C.); lee.wilson@usask.ca (L.D.W.); Tel.: +1-306-966-3244 (D.E.C.); +1-306-966-2961 (L.D.W.)

**Abstract:** The current research is focused on the mutual comparison (mechanical properties, response to humidity) of agro-waste composite materials. The purpose of this work is directed at the valorization of agro-waste biomass products and to investigate their mechanical stability for transport or other applications (in dry and wet states). Three different types of agro-waste (oat hull (Oh), torrefied wheat straw (S), and spent coffee grounds (SCG)) were blended with kaolinite (K) and chitosan (CHT) at variable weight ratios to yield ternary composites. Mechanical properties were represented by measuring hardness (in compression mode) and elastic modulus (under tension mode). Young's (elastic) modulus was measured both for dried and hydrated samples. The pelletized materials were prepared in two forms: crosslinked (CL) with epichlorohydrin and non-crosslinked (NCL). The hardness of the Oh pellets was poor (75 N) and decreased by four times with greater agro-waste content, while crosslinking affected the hardness only slightly. S pellets had the highest level of hardness at 40% agro-waste content (160 N), with a concomitant decrease to 120 N upon crosslinking. SCG pellets had the least change in hardness for both CL and NCL specimens (105–120 N). The trends of Young's modulus were similar to hardness. Hydration caused the elastic modulus to decrease ca. 100-fold. In general, S and SCG composites exhibit the greatest hardness and Young's modulus compared to Oh composites (CL or NCL) in their dry state.

**Keywords:** density; Young's modulus; chitosan; kaolinite; pellets; ternary composites; tensile tests; hardness



## 1. Introduction

Transitioning to a sustainable and circular economy is a crucial endeavor not only to decrease greenhouse gas emissions, but also to increase energy independence and enable local producers to valorize renewable materials such as agro-waste biomass [1,2]. The utilization of low-value agricultural waste for higher-value applications plays a key role in this transition and enables a variety of potential applications, from energy generation [3] to adsorbents for water purification [4,5]. Much research has been undertaken on blending polymers with biomaterials and their effect on mechanical properties [6,7]. Furthermore, oat hull (Oh) and wheat straw (S) biomass are of particular interest to the agricultural regions of western Canada, such as Saskatchewan, and other agriculturally intensive countries, such as USA, China, and Europe. Accompanying challenges and investigations in mechanical properties were reported by Agu et al. [8] Herein, a lower energy and divergent blending strategy for biomass with increased sustainability is reported, while contributing to a circular economy and biomass valorization.

To valorize such lignocellulosic materials (Oh, S, SCG) for various applications that support a circular economy, the formation of composites and other forms of physical or chemical modification offers a strategy to reach this goal [5]. In the case of wheat straw, torrefaction is particularly interesting because it serves to decrease hydrophilicity and

yield more environmentally stable materials. Furthermore, torrefaction beneficially affects the energy density and quality of pellets for energy production [8–10]. Another available agro-waste resource is spent coffee ground (SCG), which has limited utility and value after the brewing process. Converting these resources into pelletized systems for potential energy sources and/or as composite materials for water treatment can support a circular economy through the principles of green chemistry [11].

The pelletization of composites that contain agro-waste composites affects various properties, such as mechanical stability, durability, and water adsorption, according to changes in the chemical and physical properties due to compositing [8]. The focus of this study is on agro-waste from crops (unmodified oat hull, torrefied wheat straw, and spent coffee grounds) as platform materials for the design of multicomponent composites that contain kaolinite (K) and chitosan (CHT) at variable compositions. The impetus for such ternary composites is based on their facile preparation and uniquely tailored properties according to variations in the composition of the additive species [5].

Kaolinite with fixed composition and CHT with variable composition were selected as benign additives that serve as both a binder and filler. The resulting composites were investigated to explore the role of variable biomass content on the physical characteristics of the pellet materials [12–14]. The use of pellets have been established as a suitable morphology for feed and fuel products, along with pelletized adsorbents for dynamic separations in saline aqueous media [15]. An emphasis on hardness properties was made to probe storage and handling properties since pellets are oftentimes loaded in the transverse direction based on the cylindrical shape [16] of the pellets, including the role of moisture uptake at 100% humidity on the pellet properties. To address the latter, Young´s moduli were measured in addition to the volume and density changes of the pellets.

This method of pelletization represents a greener approach to pellet preparation for combustion, adsorbent technology, packaging materials, and soil amendment materials. The current method of pelletization is a non-thermal approach with a low energy footprint. In contrast to more energy-intense blending with synthetic polymers, it explores an alternate low-cost and low-energy fabrication process through room-temperature solvent-assisted blending method that allows for a broader range of feedstocks and advance the field of biomass pellet technology, ranging from fuel to adsorbent materials [8]. The materials are suitable for the adsorption of water, organics, and metal ion species. Several key articles published in recent years are cited in Table 1.

**Table 1.** Summary of various chitosan/biomass pellet-typed sorbent materials for fuel and/or water remediation.

| Adsorbent | Year | References |
|---|---|---|
| Chitosan/*Sargassum* sp. composite sorbent | 2011 | Liu et al. [17] |
| Chitosan pellets | 2017 | Mohamed et al. [18] |
| Chitosan/wheat straw pellet materials | 2020 | Mohamed et al. [19] |
| Chitosan/alginate sorbent | 2020 | Hassan et al. [20] |
| Chitosan/*E. coli* biomass sorbent | 2021 | Kim [21] |
| Chitosan/oat hull or wheat straw/kaolinite | 2022 | Mohamed et al. [22] |
| Chitosan/coconut husk pellet sorbent | 2022 | Thongsamer et al. [23] |
| Chitosan/oat hull, wheat straw or coffee/kaolinite | 2023 | Steiger et al. [5] |

## 2. Materials and Methods

### 2.1. Materials

Oat hulls were obtained from Richardson Milling Ltd. (Saskatchewan, Canada). Wheat straw was obtained in torrefied form (220 °C, 8.6 min, oxygen exclusion) from the torrefaction plant in the College of Engineering at the University of Saskatchewan. The two-stage plant consists of two horizontal screw-driven moving beds, with a single drying/preheating stage and a torrefaction stage. The plant has a throughput of 10 kg of material per hour. A full description of the plant, including the process flow diagram, is described by Camp-

bell [24]. Coffee was acquired from Real Canadian Superstore (President's Choice brand), collected, and dried at 60 °C for 2 days after coffee preparation. Kaolinite, KBr (FT-IR-grade, 99%+), and epichlorohydrin (99%+) were obtained from Sigma-Aldrich (Oakville, ON, Canada). Chitosan was provided by the Pilot Plant Corp (Deacetylation Degree: ca. 87% based on [1]H-NMR spectroscopy). Glacial acetic acid (99.7%), NaOH (97%), and HCl (conc. 37%) were obtained from Fisher Scientific, Canada. For synthesis, ultra-pure water with a resistivity of 18.2 MOhm× cm was used. All chemicals were used as received unless stated otherwise.

## 2.2. Preparation of the Composite Materials

S and Oh biomass were ground and sieved (ca. 65–67 wt.% between mesh 40–100 and 33–35 wt.% below mesh 100; Endecotts Mesh 40 with 425 μm, Endecotts Mesh 100 with 150 μm). The dried SCG was used as received [5]. The composite material (ca. 10 g total) was physically blended depending on the desired composition by weight along with 0.2 M aqueous acetic acid (ca. 20 mL) and thoroughly mixed until a pasty consistency was reached. The paste was extruded through a glass syringe (ca. 5 mm diameter) onto a paper substrate and then cut into ca. 5–7 mm lengths. The pellets were dried for 24 h at 23 °C. Then, the pellets were submerged into ca. 250 mL of 0.5 M NaOH overnight (ca. 16 h) without shaking. Afterward, the pellets were washed with Millipore water until a pH 7–8 was reached, and the pellets were subsequently dried at 295 K for 24 h. Crosslinked pellets were prepared analogous to the non-crosslinked pellets, except a 1.8 M NaOH solution (250 mL) in lieu of the 0.5 M NaOH solution. The addition of epichlorohydrin (1.12 wt.% ECH Solution, Gloucestershire, UK; 2.8 mL) to the 1.8 M NaOH solution [25,26] was followed by light shaking at 180 rpm (Scilogex SK-0330-Pro, Scilogex, Rocky Hill, CT, USA) overnight. The increase in NaOH was aimed at allowing efficient crosslinking at lower ECH concentrations, as reported elsewhere [27]. After 16–18 h of crosslinking, the pellets were washed with water until a pH of 7–8 was obtained, followed by drying at room temperature. Crosslinked pellets are referred to with the designation of the pellet composition and of the suffix "-CL" (e.g., Oh20-CL), based on the sample ID naming convention in Table 2. For example, an Agro-waste 20 composite contains 20 wt.% of either oat hulls (e.g., Oh composite), wheat straw (e.g., S composite), or spent coffee ground (e.g., SCG composite) with chitosan and kaolinite at 70 and 10 wt.%, respectively.

**Table 2.** Agro-waste biomass (oat hulls, Oh; wheat straw, S; or spent coffee ground, SCG), binder (kaolinite), and filler (chitosan) content by weight (%) for the three types of composites.

|  | Agro-Waste 20 | Agro-Waste 40 | Agro-Waste 60 | Agro-Waste 80 |
|---|---|---|---|---|
| Oh composites | 20 | 40 | 60 | 80 * |
| S composites | 20 | 40 | 60 | 80 |
| SCG composites | 20 | 40 | 60 | 80 |
| Chitosan | 70 | 50 | 30 | 10 |
| Kaolinite | 10 | 10 | 10 | 10 |

* Note: unstable without crosslinking.

The precursor materials are shown in Figure 1.

## 2.3. Density

For the density calculations, the physical dimensions of the pellets were measured before and after exposure to humidity. The pellets were used in the form of an elongated cylinder, and a digital caliper (error ± 0.005 mm) was used to determine the dimensions. The weight was measured by a digital balance (±0.1 mg). To account for non-uniformity of the pellets, six measurements along their length and diameter were used to estimate average dimensional values.

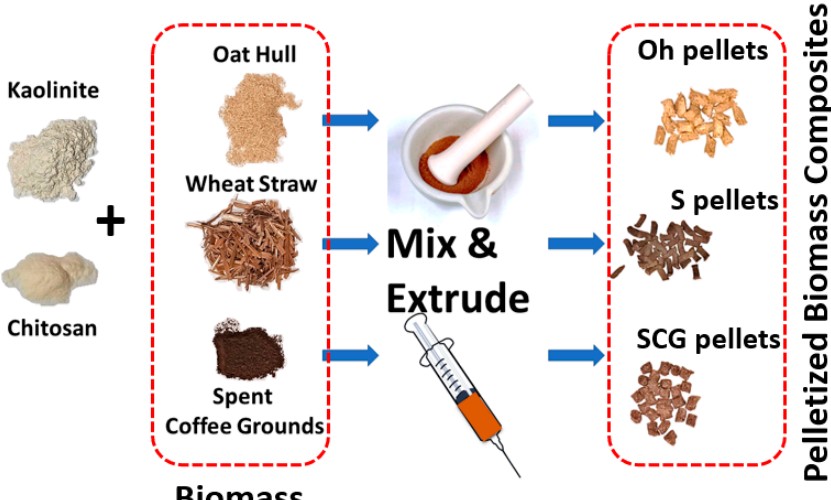

**Figure 1.** Images of the starting materials: kaolinite, chitosan, oat hulls, torrefied wheat straw and spent coffee grounds (used as received) before grinding, along with a schematic view of the pellet preparation process.

### 2.4. Moisture Content and Uptake

To measure the moisture content of the dried pellets, thermogravimetric analysis (TGA) was employed. The weight loss profiles were obtained using a Q50 TA Instruments thermogravimetric analyzer (TA Instruments, New Castle, DE, USA). Samples were heated in open aluminum pans at 30 °C for 5 min to allow for equilibration prior to heating at 10 °C/min to 200 °C. A humidity chamber with saturated $K_2SO_4$ solution was used to achieve an RH of 97%. The samples were placed for 96 h in the humidity chamber, and moisture uptake was calculated after they reached a constant weight. Two modes were used for assessing their moisture uptake, by mass and by volume. The water uptake was determined as a ratio of *final value–initial value/initial value* and expressed as wt.% accordingly, where each measurement was obtained in triplicate.

### 2.5. SEM Imaging

The surface morphology of the samples was surveyed by a Jeol JSM-6010LV (Tokyo, Japan) scanning electron microscope (SEM) with a tungsten filament source at an accelerated voltage of 10 kV. The instrument was operated under a high vacuum imaging mode using the secondary electron image (SEI) detector. The composites were gold coated at high vacuum prior to SEM interrogation using an Edwards S150B sputter coater (Crawley, West Sussex, UK). All images were taken in slow scan mode (60 s$^{-1}$).

### 2.6. Mechanical Tests

#### 2.6.1. Hardness

Hardness was evaluated by cutting the pellets into ca. 3 mm thick tablets, according to an adapted procedure described elsewhere [28,29], and a single pellet was placed on its side between two flat platens. The load was applied diametrically on the portions of the curved surfaces of the tablets by applying a constant load at a rate of 0.2 mm/s using a Mark-10 Force Test Stand, ESM 1500LC (Copiague, NY, USA), equipped with a 500 N load cell. The hardness was measured in a compression mode by taking the maximum load to cause fracture resulting in cracking or breaking of the pellets [30]. This load was taken as a measure of pellet hardness [31]. Five pellets for each system were used for the average hardness.

#### 2.6.2. Young's Modulus

To evaluate Young's modulus of the composite pellets, tension tests were performed. The samples were subjected to a controlled tension until their failure. All measurements

were conducted using a Mark-10 Force Test Stand, ESM 1500LC, according to the guidance of ASTM D613-14 standard for testing plastics under tension.

The instrument was equipped with a 500 N load cell and a Mark-10 Model 5i digital force indicator. The specimens were fixed between two tensile parallel jaw grips, large-model G1100. Young's (or elastic) modulus was determined for the test specimens and calculated from the slope of the linear portion of the stress–strain curve. The measurement for each sample was obtained in triplicate, and the standard deviations, along with the absolute errors, were calculated. Young's modulus is a constant for a particular material, and its value is obtained from the stress–strain ratio when the material behaves in a linear-elastic way. Stress is defined as the ratio of the applied load (*F*) to the original cross-sectional area (*S*) of a sample (*F/S*); and strain is the relative elongation ($\Delta l/l$) of a material, where *l* is the original length of the specimen. Thus, Young's modulus is expressed as follows (Equation (1)):

$$E = \frac{Fl}{S\Delta l} \qquad (1)$$

All physical quantities are expressed in SI units, with the subsequent conversion of *E* to MPa.

## 3. Results

As outlined in the Introduction, several types of ternary composites were prepared with variable compositions of agro-waste biomass, kaolinite, and chitosan (cf. Table 2). The corresponding materials were prepared with and without crosslinking, along with the characterization of the morphology and mechanical properties such as hardness and Young's modulus, density, and moisture uptake. Scanning electron micrograph imaging was obtained to view the surface morphology and to correlate the structure–function relationships of the various pellet materials. The physicomechanical tests yielded such properties as hardness and elastic modulus, along with density and moisture uptake. The results show how hydration and crosslinking affect the mechanical properties of the pellets and their correlation with the density and moisture uptake, which are relevant for various technical applications of such composites.

### 3.1. SEM Imaging

Herein, the surface morphology and textural properties of the composites were evaluated via SEM imaging (cf. Figure 2) according to the changes in structure upon crosslinking. In the case of biopolymers, crosslinking is known to affect the surface chemistry and site accessibility of functional groups, according to steric effects, along with the textural properties of the resulting material [32,33].

In general, the differences between the more fibrous agro-waste (Oh, S) and granular SCG are visible via the SEM images. The surface features of the composites without crosslinking (NCL) are evidenced for the pellet systems that contain oat hulls, where the presence of heterogeneous particles/fibers of the biomass is most evident.

Oh composites show a rather coarse agro-waste distribution in their non-crosslinked form (cf. Figure 2A,B), while crosslinking (cf. Figure 2C,D) appeared to degrade the structure due to the NaOH (*aq*) media employed. Torrefied wheat straw pellets (Figure 2E,H) show a more elongated, fibrous structure in comparison, which is more evident for the crosslinked materials (Figure 2G,H). Crosslinking via ECH does not show significant changes for the S-based composite materials, which indicates a more stable composite. SCG composites show a more pronounced alteration of the surface structure upon crosslinking.

Crosslinked SCG pellets (Figure 2L,M) show a different surface morphology than NCL SCG (Figure 2J,K), as evidenced by additional surface features that resemble small ridges [34]. However, this change in structure and surface properties may relate to more efficient washing and removal of soluble species such as lignins, as evidenced by a deeper colored filtrate observed during the NaOH washing rather than induced by ECH.

It can be purported that the difference in structure is based on the physical form of the biomass fraction (Oh, SCG, S), where interactions with the chitosan–kaolinite matrix affect the nature of the pellets alongside their chemical properties (hydrophilicity, H-bonding, density).

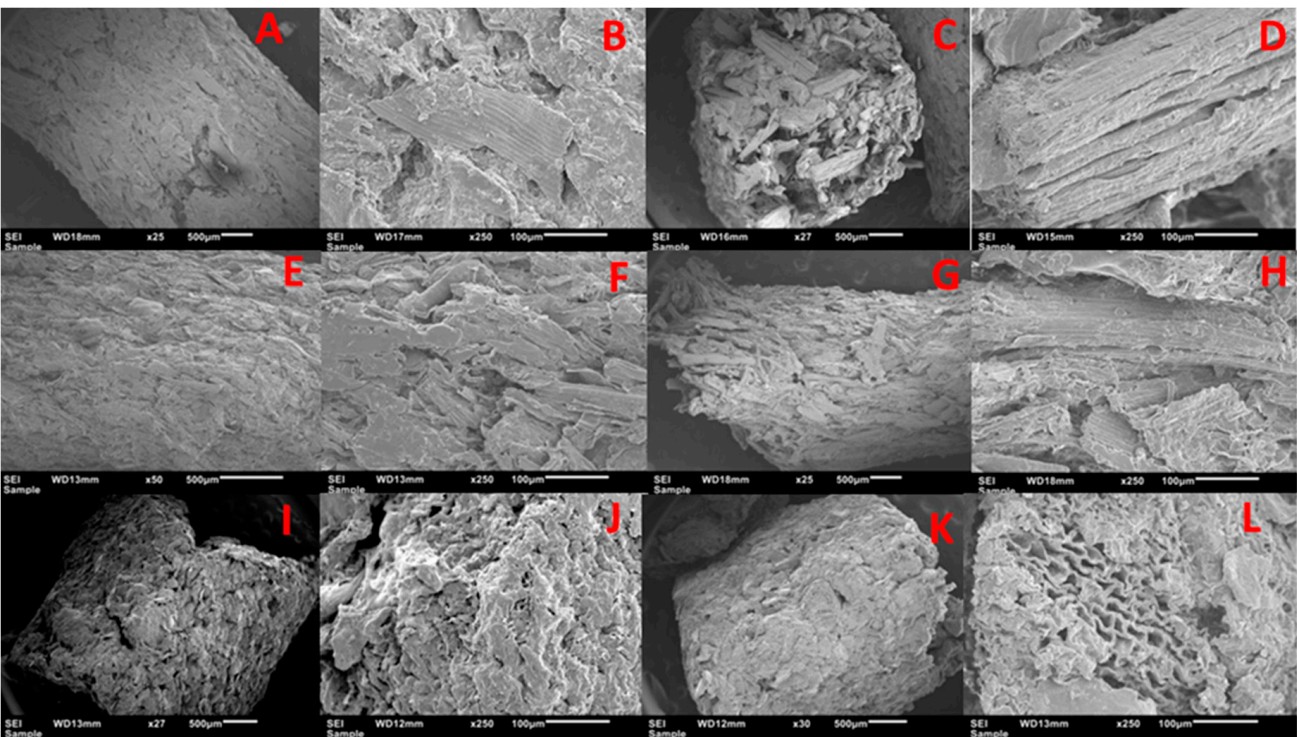

**Figure 2.** Low magnification images for 60 wt.% pellet materials with oat hulls (**A,B**), crosslinked oat hull pellets (**C,D**), torrefied wheat straw (**E,F**), crosslinked wheat straw pellets (**G,H**), spent coffee grounds (**I,J**), and crosslinked spent coffee pellets (**K,L**).

### 3.2. Hardness

Herein, the results for the hardness measurements provide insight as to how the composites endure storage and transportation. If the load exerted onto the pellets along the vertical axis exceeds the hardness in Newtons (N), breaking and crumbling will occur. Packaging and handling of the composite pellets, therefore, must take the hardness into account for preserving the composite structural integrity. In general, oat hull pellets were more likely to break apart as expected, whereas the other pellet systems exhibit a more plasticized nature (microfracturing). Herein, a chitosan binary pellet system with 10% K and 90% CHT was tested as a baseline to compare the effects of blending and crosslinking (Figure 3). The large error bars may be attributed to the process of manually cutting the pellets and heterogeneity across the samples.

In general, Oh shows the lowest hardness of all pellet systems, where the hardness decreases significantly (from 76 N to 19 N) with incremental agro-waste content. Crosslinking aids in preserving the hardness (from 72 N to 58 N) for 20–60% Oh content, while Oh80-CL exhibits the lowest stability (26 N), slightly greater than Oh60 without crosslinking.

By contrast, S-based agro-waste pellets show an improved hardness compared to chitosan–kaolinite pellets, where an improvement in hardness (161 N) occurs with an increase in agro-waste content. Upon reaching 80% S content, the hardness (83 N) was comparable to the hardest Oh-systems. Crosslinking led to decreased hardness when compared to non-crosslinked pellets, which indicates that the structural integrity becomes compromised during the conditions employed in the course of chemical modification.

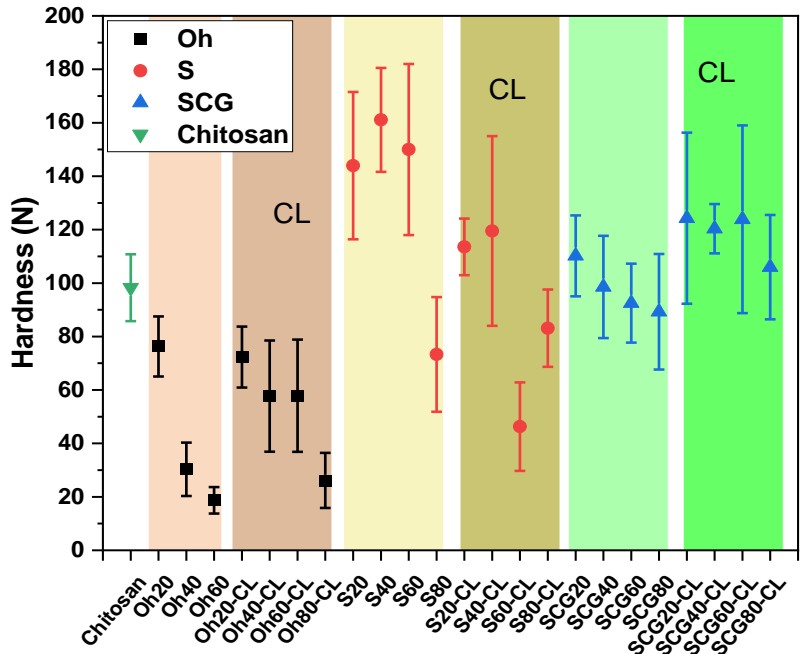

**Figure 3.** Hardness of the Oh-, S-, and SCG-pellet systems compared to 90% CHT-pellets.

SCG pellets are contrasted with the Oh and S composites, which show the least spread in hardness (constant decrease from 110 N to 89 N with an increase in agro-waste content for non-crosslinked and from 124 N to 106 N for the crosslinked composites). Crosslinking slightly improves the hardness for the SCG composites.

Pellet formation and binding characteristics can be beneficially influenced through added heat and temperature (with associated high energy input depending on the specific method) [35]. The strategy outlined herein avoids additional energy input by obviating added pressure and heat while aiming for comparable characteristics. Tilay et al. described that reasonable durability of canola-derived fuel pellets observed a hardness of 71 N (max. 189 N), whereas poor quality pellets were indicated by ca. 30 N hardness [29]. Therefore, SCG composites and S20–S60 (crosslinked or non-crosslinked) possess acceptable hardness, whereas Oh composites above 40 wt.% content show poor quality. It should be pointed out that the method reported by Tilay et al. [29] versus the method employed herein differs significantly due to the lack of heating in the current method and its overall sustainability, according to the more mild ambient conditions employed in Section 2.2.

### 3.3. Modulus of Elasticity in Tension (Young's Modulus)

The formation of composites as a physical blend and pelletized form are expected to be exposed to various forms of stress in their dry state for transport and storage applications. For the case of adsorbents in solid–liquid systems, an evaluation of stress in their wet forms by using Young´s moduli (E) can provide insight for practical applications. Firstly, the dry composites were evaluated to obtain a baseline for stability under storage and transport conditions, which were then compared against the wet materials (cf. Table 3).

For both SCG and S composites without crosslinking, a biomass content of 40 and 60% appears to show the highest Young's modulus, as compared to lower or higher biomass content. However, Oh composites exhibit their highest elastic modulus with the lowest biomass (Oh20) content.

**Table 3.** Young's moduli of dry chitosan-based agro-waste pellets in their crosslinked (CL) and non-crosslinked (NCL) forms in the dry state.

| Biomass wt.% | Coffee Ground | | Oat Hull | | Wheat Straw | |
|:---:|:---:|:---:|:---:|:---:|:---:|:---:|
| | NCL | CL | NCL | CL | NCL | CL |
| | E, MPa | | E, MPa | | E, MPa | |
| 0 | 816 ± 44 * | | | | 629 ± 29 ** | |
| 20 | 662 ± 36 | 842 ± 49 | 791 ± 43 | 887 ± 50 | 1245 ± 75 | 1023 ± 55 |
| 40 | 779 ± 31 | 954 ± 47 | 597 ± 33 | 748 ± 46 | 1447 ± 52 | 939 ± 54 |
| 60 | 674 ± 39 | 348 ± 26 | 344 ± 30 | 519 ± 27 | 1447 ± 64 | 885 ± 44 |
| 80 | 506 ± 29 | 519 ± 38 | - | 401 ± 31 | 698 ± 42 | 939 ± 51 |

* 0 wt.% content of biomass refers to pelletized chitosan (CHT) in its non-crosslinked form as a reference; ** 0% biomass, 90% CHT, and 10% kaolinite (*w/w* content) *w/o* crosslinking for comparison.

Upon crosslinking, the Oh composites show a consistently greater elastic modulus when compared to the non-crosslinked composites, whereas no trends were evident for either SCG or S composites (cf. Figure 2). It is posited that a greater concentration of NaOH during the crosslinking process led to greater leaching of soluble components (e.g., lignins, hemicellulose) from the straw and coffee biomass, in addition to increased degradation of the glycosidic constituents, as reported elsewhere [5]. For the case of the Oh composites, the greater -OH content and availability of this biomass may offset degradation by allowing for more efficient crosslinking since the Oh was not torrefied (unlike S and SCG). However, these results were obtained in the dry state, without any hydrostatic pressure. In contrast with Table 3, the results in Table 4 illustrate the role of hydration and swelling according to Young's moduli.

**Table 4.** Young's moduli of hydrated chitosan-based biomass pellets in their crosslinked (CL) and non-crosslinked (NCL) forms.

| wt.% | Coffee Ground | | Oat Hull | | Wheat Straw | |
|:---:|:---:|:---:|:---:|:---:|:---:|:---:|
| | NCL | CL | NCL | CL | NCL | CL |
| | E, MPa | | E, MPa | | E, MPa | |
| 0 | 95 ± 10 * | | | | 86 ± 8 ** | |
| 20 | 37 ± 4 | 169 ± 12 | 202 ± 15 | 275 ± 19 | 161 ± 13 | 180 ± 17 |
| 40 | 26 ± 3 | 87 ± 9 | 67 ± 6 | 236 ± 16 | 127 ± 12 | 169 ± 13 |
| 60 | 25 ± 3 | 37 ± 3 | 11 ± 1 | 201 ± 11 | 102 ± 10 | 142 ± 14 |
| 80 | 13 ± 1 | 15 ± 1 | - | 149 ± 11 | 50 ± 4 | 122 ± 11 |

* 0 wt.% content of biomass refers to pelletized chitosan (CHT) in its non-crosslinked form as a reference; ** 0% biomass, 90% CHT, and 10% kaolinite (*w/w* content) *w/o* crosslinking as comparison.

To gain insight into the role of hydration of the composite pellets and to gauge the change in the mechanical properties and stability at relevant conditions, Young's moduli of the hydrated composites and their properties are presented at variable biomass content (cf. Table 4 and Figure 4).

The hydrated samples are shown to exhibit up to a 100-fold reduction in Young's modulus as compared to their dry analogs (cf. Table 3). For the case of S composites without crosslinking, such materials are nearly five-fold stiffer than Oh and SCG composites at the highest biomass content (cf. Figure 4).

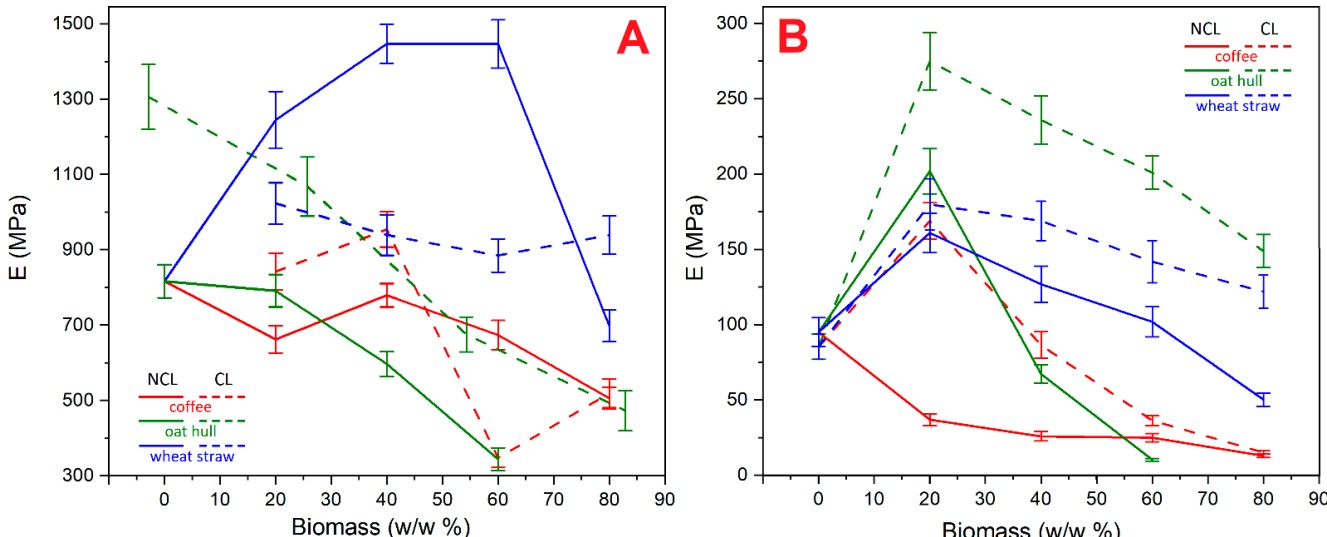

**Figure 4.** Comparative charts of Young's moduli (E) vs. biomass content (%) for the crosslinked (CL) and non-crosslinked (NCL) composite pellets in the dry (**A**) and hydrated (**B**) states.

In contrast to the variation in Young's modulus of pellets in their dry state (cf. Table 3), crosslinking improved elastic modulus over their NCL counterparts after hydration (cf. Table 4). A uniform trend observed across all materials was a decrease in Young's modulus, which was correlated with greater biomass content. Oat hull-based composites (NCL) showed a tremendous (~20-fold) decrease in their elastic modulus as the mass fraction of biomass content increased from 20 to 60%. Remarkably, both CL and NCL SCG composites exhibit an N-shaped trend, where pellets with 40% of SCG have the highest Young's modulus. This could be explained by specific chemical interactions between coffee particles and chitosan at moderate SCG content, whereas higher SCG content leads to the breakage of these bonds with the subsequent decrease in their mechanical strength. In addition, this refers to the lower CHT content, which acts as an adhesive agent due to its amphiphilic interactions with both biomass and kaolinite. By comparison, CL oat hull composites did not exhibit such an abrupt change in Young's modulus. SCG-based pellets had the lowest elastic modulus in the hydrated state. The introduction of water and hydration interactions (H-bonding) with the additive components of the composites may attenuate the H-bonding and other adhesive interactions between the solid components (mostly cellulosic materials), which concur with a decrease in the mechanical strength [36].

### 3.4. Density

An increase in the hydration of a composite can influence the density of the materials, as shown by the trends in Table 5.

**Table 5.** Densities ($\rho$) of dry agro-waste pellets with crosslinking (CL) and without crosslinking (NCL).

| wt.% | Coffee Ground NCL | Coffee Ground CL | Oat Hull NCL | Oat Hull CL | Wheat Straw NCL | Wheat Straw CL |
|---|---|---|---|---|---|---|
| | $\rho$, g/cm$^3$ | | $\rho$, g/cm$^3$ | | $\rho$, g/cm$^3$ | |
| **0** | 0.88 ± 0.03 * | | | | 0.96 ± 0.03 ** | |
| **20** | 0.99 ± 0.02 | 1.13 ± 0.01 | 0.98 ± 0.03 | 1.02 ± 0.01 | 1.07 ± 0.06 | 1.05 ± 0.03 |
| **40** | 1.06 ± 0.04 | 1.16 ± 0.03 | 0.88 ± 0.03 | 1.00 ± 0.02 | 1.24 ± 0.05 | 1.08 ± 0.01 |
| **60** | 1.10 ± 0.03 | 1.20 ± 0.02 | 0.80 ± 0.05 | 0.98 ± 0.03 | 1.29 ± 0.03 | 1.13 ± 0.02 |
| **80** | 1.13 ± 0.01 | 1.22 ± 0.01 | - | 0.92 ± 0.05 | 0.86 ± 0.03 | 1.00 ± 0.02 |

* 0 wt.% content of biomass stands for pure CHT in its NCL form as comparison. ** 0 wt.% biomass, 90% CHT, and 10% kaolinite (wt.% content) NCL for comparison.

The chitosan pellets have a lower density than the biomass pellets, which indicates the greater porosity of chitosan pellets, in agreement with another reported study [37]. SCG-based composites showed a gradual increase in density from SCG20-SCG80, whereas the Oh and S composites achieved the highest density at 40–60 wt.% biomass. Beyond these upper limits, the density decreases in a parallel manner, along with the optimal mechanical properties (tensile strength/stiffness/elastic modulus). The density of kaolinite-modified chitosan is slightly higher (0.96 vs. 0.88 g/cm$^3$) versus pure chitosan pellets [38,39]. As noted in Figure 5A, the S composites tend to achieve the highest density, which approaches 1.3 g/cm$^3$. The density of a single pellet falls into the range of 0.9–1.3 g/cm$^3$, which is in good agreement with a study reported by Agu et al. (1.0–1.4 g/cm$^3$) [8].

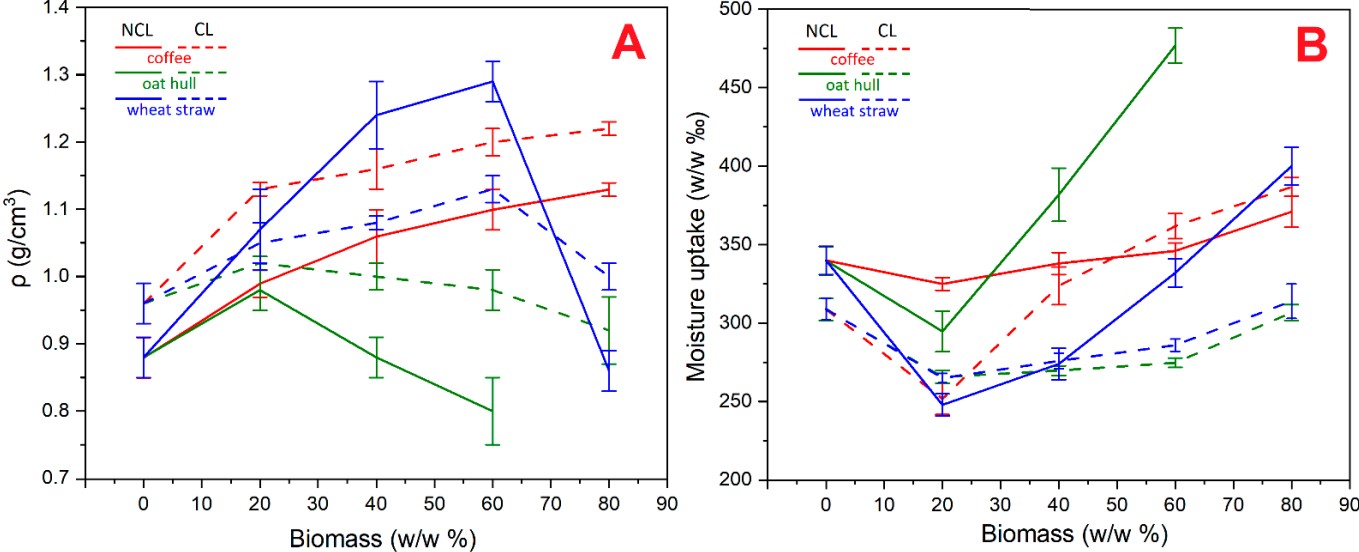

**Figure 5.** Comparative charts of density ($\rho$) vs. biomass content (%) (**A**), and a comparison of vapor uptake (*w/w*) vs. biomass content (**B**) for crosslinked (CL) and non-crosslinked (NCL) composite materials in their dry state.

Crosslinking with ECH provides certain mechanical reinforcement (*vide supra*) for the composites, which is reflected in the incremental density values. It was expected that CL composites show a higher density, where this trend is not obeyed for the SCG60 and SCG80 composites. This implies that unique chemical interactions occur for these composites. S composites showed a decrease in density upon crosslinking, which parallels their improved mechanical properties (except S80; cf. Table 4).

### 3.5. Moisture Uptake

To establish a better understanding of the effects of moisture uptake on the pellets, the moisture content at 23 °C and laboratory conditions was evaluated (cf. Table 6 and Figure 6).

**Table 6.** Water content at 23 °C and laboratory conditions at ambient humidity calculated through the weight loss at 150 °C.

|  | Agro-Waste (20%) | Agro-Waste (40%) | Agro-Waste (60%) | Agro-Waste (80%) |
|---|---|---|---|---|
| Oh composites | 6.5 ± 0.2 | 6.4 ± 0.2 | 6.1 ± 0.2 | N/A |
| Oh composites (CL) | 6.5 ± 0.2 | 6.6 ± 0.2 | 3.1 ± 0.2 | 5.7 ± 0.2 |
| S composites | 6.5 ± 0.2 | 6.9 ± 0.2 | 6.0 ± 0.2 | 5.3 ± 0.2 |
| S composites (CL) | 6.5 ± 0.2 | 6.9 ± 0.2 | 4.8 ± 0.2 | 5.9 ± 0.2 |
| SCG composites | 6.5 ± 0.2 | 6.9 ± 0.2 | 6.5 ± 0.2 | 6.2 ± 0.2 |
| SCG composites (CL) | 6.5 ± 0.2 | 6.4 ± 0.2 | 6.3 ± 0.2 | 5.9 ± 0.2 |

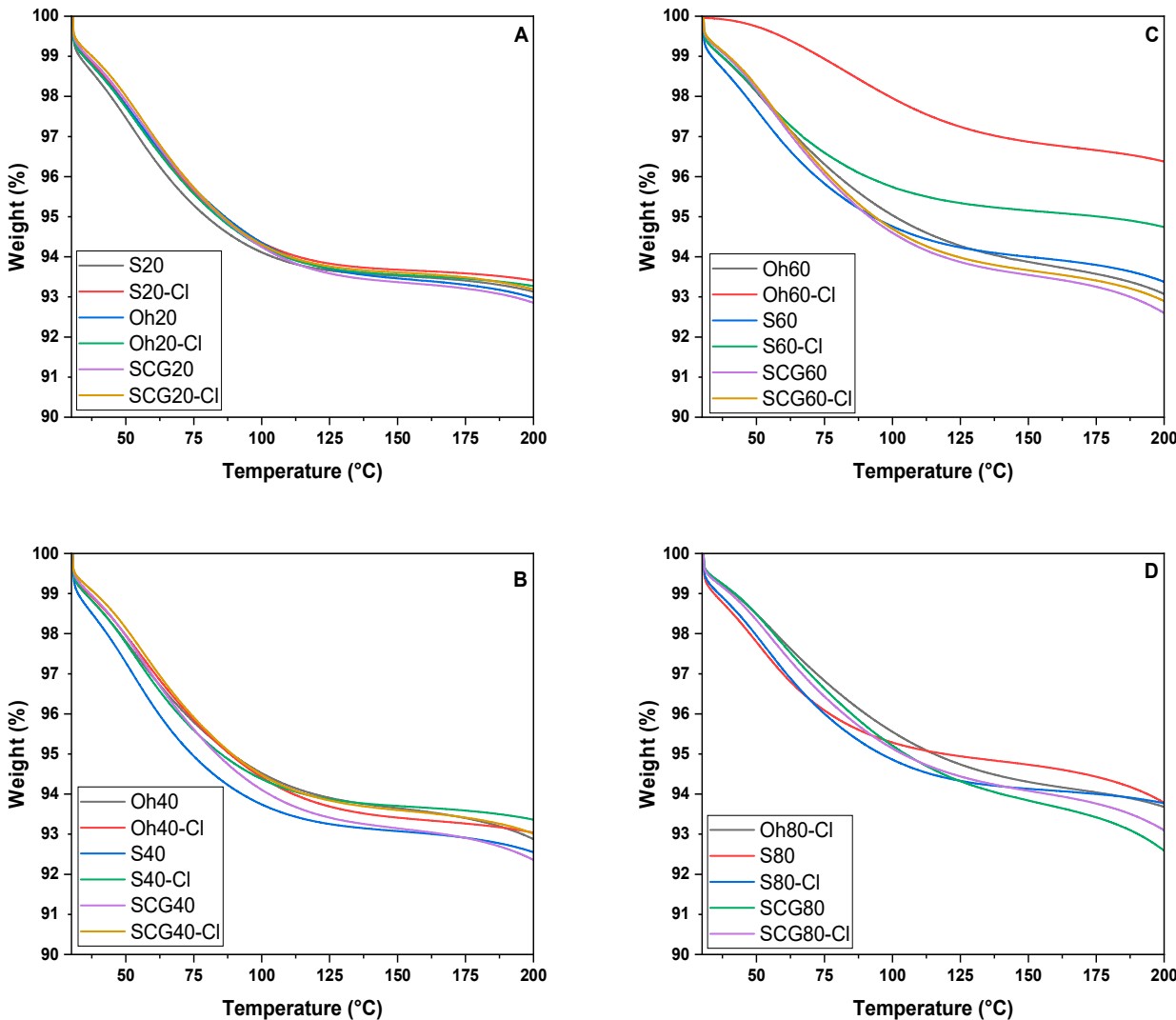

**Figure 6.** Estimation of water content of dried composites through TGA decomposition profiles, where each panel (A-D) describes composites with variable biomass content: (**A**) 20 wt.%, (**B**) 40 wt.%, (**C**) 60 wt. % and (**D**) 80 wt.%.

At low biomass content, there is no appreciable difference between the various composites (ca. 6.5 % for 20% agro-waste content). When the biomass content was increased to 60%, the NCL composites had similar moisture content near 6.0–6.5%, but the CL Oh and S composites had decreased water contents (3.1% and 4.8%, respectively). Crosslinking did not appreciably affect SCG composites. For 80% agro-waste content, no apparent trend was observed, unlike composites with 60% biomass.

Starting from ca. 5–7% water content in the dry composites, the density and moisture uptake (i.e., weight loss) can be measured to assess the relative hydrophilicity of the composite materials and to infer the stability of the composites in water. The moisture uptake (cf. Table 7 and Figure 5) for SCG, Oh, and S composites normally vary from 300 to 400‰. However, the Oh composites showed the greatest water uptake (up to 477 ‰), which concurs with the greater hydrophilic character of oat hulls (non-torrefied), in contrast to S and SCG agro-waste composites.

**Table 7.** Moisture uptake by chitosan-based pellets with (CL) and without crosslinking (NCL) at RH 97%.

| wt.% | Spent Coffee Grounds | | Oat Hulls | | Wheat Straw | |
| | NCL | CL | NCL | CL | NCL | CL |
| | Moisture Uptake, ‰ (*w/w*) | | Moisture Uptake, ‰ (*w/w*) | | Moisture Uptake, ‰ (*w/w*) | |
|---|---|---|---|---|---|---|
| **0** | 340 ± 9 * | | | | 309 ± 7 ** | |
| **20** | 325 ± 4 | 252 ± 10 | 295 ± 13 | 266 ± 4 | 248 ± 7 | 265 ± 3 |
| **40** | 338 ± 7 | 324 ± 12 | 382 ± 17 | 270 ± 3 | 274 ± 10 | 276 ± 5 |
| **60** | 346 ± 5 | 362 ± 8 | 477 ± 11 | 275 ± 3 | 332 ± 9 | 286 ± 4 |
| **80** | 371 ± 10 | 387 ± 6 | - | 307 ± 5 | 400 ± 12 | 314 ± 11 |

* 0 wt.% content of biomass stands for pure CHT in its non-crosslinked form as comparison. ** 0% biomass, 90% CHT and 10% kaolinite (*w/w* content) NCL as comparison.

A higher moisture uptake was expected for the Oh composites, as compared with the SCG and S composites; since the Oh biomass was not torrefied, a greater content of surface hydroxy groups is present in contrast to torrefied biomass. Additionally, the relative porosity of the pellet matrix plays a role. Denser pellets may have less water content than pellets with reduced density (greater porosity). This expectation concurs with the data from Table 4 (Oh20 as an exception) and SEM images (cf. Figure 2), where Oh composites possess greater porosity. Oat hulls are more hydrophilic in nature due to lignin and hemi-cellulose content, as evidenced by their lower stability in water [5]. The interactions of the composites with water are driven by hydrophilic interactions, which are also likely to affect the adsorption properties of different pollutants according to their hydration properties. This hypothesis was supported by the case of Pb(II) adsorption with oat hull composites (40% biomass with 10% kaolinite + 50% chitosan). Oh composites display greater lead uptake than the analogous torrefied wheat straw composite, in accordance with the greater hydrophilic character and lignin content of oat hulls over wheat straw biomass [22].

*3.6. Proposed Structure and Cohesion within the Material*

Torrefied wheat straw pellets (cf. Figure 2E–H) show a more elongated, fibrous structure in comparison to crosslinked materials (Figure 2G–H). Crosslinking via ECH does not show significant changes for Oh and S composite materials, whereas SCG composites (Figure 2K,M) show slight differences.

Upon crosslinking, the SCG pellets (Figure 2L–M) show a different surface morphology than NCL (Figure 2J–K), as evidenced by additional surface features that resemble small ridges [34]. However, this change in structure and surface properties may relate to more efficient washing and removal of soluble species such as lignins, as evidenced by a deeper color in the filtrate after the NaOH washing rather than structural factors induced by ECH crosslinking.

It can be purported that the difference in structure is based on the physical form of the additive (Oh, S, SCG) and the adhesive interactions with the chitosan–kaolinite matrix (cf. Figures 3 and 4). Adhesive interactions (polar and apolar) affect the nature of the pellets alongside their physicochemical properties, such as hydrophilicity, H-bonding, and density. Figure 7 displays a schematic illustration assuming referral to NCL pelletized materials, along with Tables 4 and 5. The morphologies in Figure 7 are supposed to be inferred from the relative visual density of the packing among constituent additives. The notion of variable adhesive interactions can be inferred from the relative polarity of the constituents, along with the morphology of the various additives. In the case of torrefied wheat straw, the fibrous morphology of the biomass is complementary to CHT, where the amphiphilic character of CHT can further stabilize the kaolinite in the matrix. By comparison, Oh is a more hydrophilic biomass that can interact favorably with chitosan and kaolinite, however; the fibrous structure is less extensive than S-based biomass, which reduces the effective surface interactions of this system. The granular morphology of SCG has a lower aspect ratio than that of the fibrous constituents of S-based composites, which results in reduced

adhesive interactions for this pelletized system. It should be noted that SCG and S biomass differs from Oh due to the relative polar and apolar character of the biomass, which can be offset according to incompatibility (defects) between components.

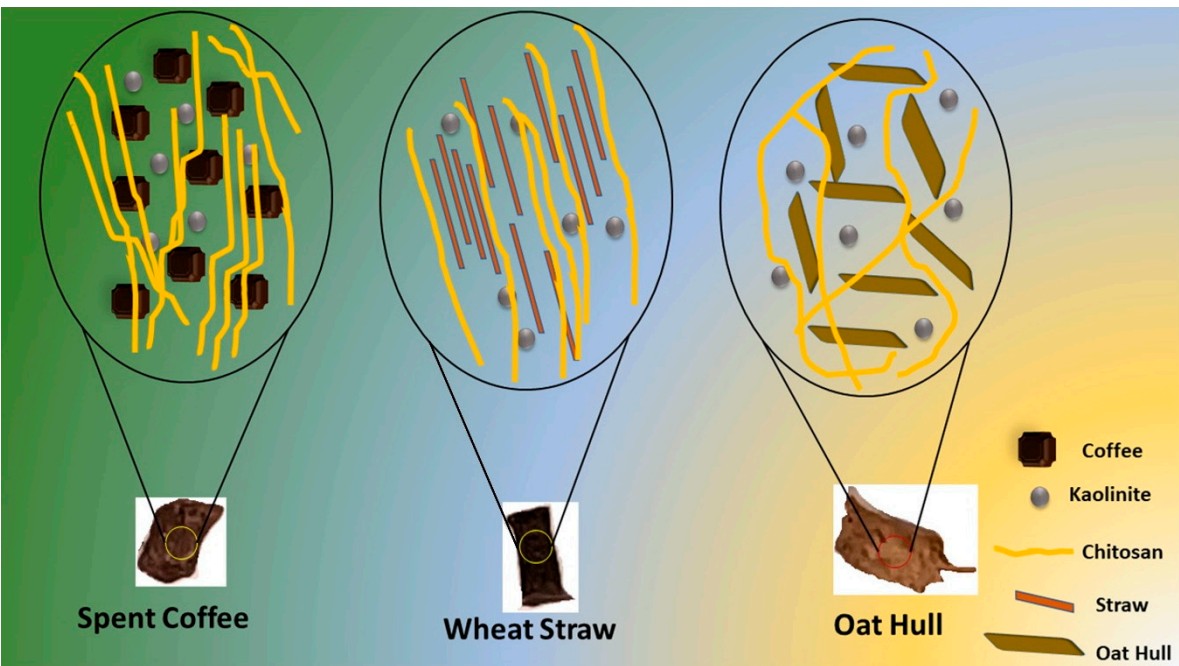

**Figure 7.** Schematic illustration of the pelletized NCL composites that conceptually illustrate different morphologies and densities between biomass particles (SCG, S, and Oh) within a matrix containing chitosan and kaolinite.

## 4. Conclusions

Three different pelletized composites with variable content of agro-waste (Oh, S, and SCG) were prepared, where the physicochemical and mechanical properties were investigated. For Oh composites, the hardness decreased with increasing biomass content (from 73 N to 19 N), whereas crosslinking deteriorated the pellet hardness (from 72 N to 26 N). S composites significantly outperformed the Oh NCL composites (161 N to 73 N), while CL decreased the maximum hardness (119 N to 46 N). SCG composites were much less affected by composition (110 N to 89 N); even with crosslinking, a minor increase in hardness could be observed (124 N to 106 N). Young´s moduli (cf. Tables 3 and 4) showed that chitosan/kaolinite binary composite material displayed lower Young´s moduli (629 MPa) compared to pure chitosan pellets (816 MPa) in the dry state. Blending torrefied wheat straw could increase Young´s modulus up to 1447 MPa (S40, S60), whereas Oh composites showed an inverse relationship between biomass content and stress (Oh80 was unstable in water). In certain cases, crosslinking increased (S80, Oh60, SCG20-40) or decreased (S20-60, SCG60) the mechanical strength. Most importantly, upon hydration, the mechanical strength drastically decreased up to 100-fold, which explains the decomposition of Oh pellets in solution when exceeding the 60 wt.% biomass content. Here, crosslinking generally increased Young´s modulus, where an inverse relationship between elastic modulus and biomass content (the less biomass, the stronger, cf. Table 4) occurs in all cases. Upon hydration, the SCG materials showed the lowest Young´s modulus overall. In summary, Oh in an unmodified form (CL or NCL) showed significantly lower hardness and mechanical properties compared to S and SCG composites. Crosslinked SCG composites outperform S composites even at 40% and 60% of biomass content in their wet state. The pellets produced in this study are potential candidates for fuel and/or water remediation applications. Future research may improve the mechanical and physicochemical properties

by investigating the role of altering the chemical surface attributes, such as hydrophobicity, prior to pelletization.

**Author Contributions:** Conceptualization, L.D.W. and D.E.C.; methodology, L.D.W. and D.E.C.; formal analysis, Y.A.A. and B.G.K.S.; investigation, Y.A.A. and B.G.K.S.; resources, L.D.W. and D.E.C.; data curation, L.D.W. and D.E.C.; writing—original draft preparation, Y.A.A. and B.G.K.S.; writing—review and editing, Y.A.A., B.G.K.S., L.D.W. and D.E.C.; data visualization, Y.A.A. and B.G.K.S.; supervision, L.D.W. and D.E.C.; funding acquisition, L.D.W. and D.E.C. All authors have read and agreed to the published version of the manuscript.

**Funding:** This research was funded by the Government of Canada through the Natural Sciences and Engineering Research Council of Canada (Discovery Grant Number: RGPIN 04315-2021). The APC was funded by the Journal of Composites Science editorial office.

**Data Availability Statement:** Not applicable.

**Acknowledgments:** The authors are grateful to Richard W. Evitts for using his laboratory for humidity measurements and provision of the torrefied wheat straw.

**Conflicts of Interest:** The authors declare no conflict of interest.

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
