# Peer review of "Moisture Content and Mechanical Properties of Bio-Waste Pellets for Fuel and/or Water Remediation Applications"

_jcs, doi:10.3390/jcs7030100_

Round 1

Reviewer 1 Report

1 Table 1 should be redesigned, the content is not clear at a glance.

2 When testing the hardness and Young modulus, is there any standard, such as ASTM or ISO standard, which can be used? If yes, please cite the standard.

3 From the error bar shown in Figure 3, it can be found that, the standard error values for some composites are significant, how to explain this phenomenon?

4 It is suggested that, all the number of digits after the decimal point should kept the same for all the test results shown in the Tables.

5 line 288, “In contrast to the variation in the mechanical strength of pellets “, please confirm that only modulus was given in Table 2, not strength.

6 in Figure 4, the modulus of SCG-based pellets(both CL and non-CL)in the dry state, varied with the content of the biomass in the form of “N” trend, how to explain it?

7 about “moisture uptake”, only the phenomenon was marked. Lack of scientific discussion is there. 

Reviewer 2 Report

I think this manuscript is well-prepared and can be published in the Journal of Composites Science.

Reviewer 3 Report

The work titled “Moisture content and mechanical properties of bio-waste pellets for fuel and/or water remediation applications”, is an interesting research work in the field of polymer composites. Following corrections needs to be done before accepting the work.

1. The abstract of the work needs more of results and percentage improvement in the combinations.

2. At the end of introduction part novelty of this particular work needs to be mentioned more precisely.

3. This work is related to bio waste applications to different fields. See the following works related to that, add that with literature

DOI: 10.1002/pc.26362  

DOI: 10.1177/15280837211022614  

4. Why did you use bio waste for the particular application. Moisture absorption and its properties will be lower in comparison with the available material in the industry. Explain that in detail.

5. The results and discussion part need to be explained more with comparison of latest works.

6. Have you tried something to decrease the hydrophilic property of bio waste pellets.

6. I feel conclusion part is more, reduce the content with most important findings with future works.

Round 2

Reviewer 1 Report

The manuscript has been revised well according to the review report, it was suggested to be accepted by JCS.

Reviewer 3 Report

The research work can be accepted in the current form.